# The Impact of Radioactive Iodine on Outcomes Among Pediatric and Adolescent Thyroid Cancer Patients: A SEER Database Analysis

**DOI:** 10.3390/cancers17010107

**Published:** 2025-01-01

**Authors:** Emily M. Persons, Mohammad H. Hussein, Marcela Herrera, Dylan Pinion, Alyssa Webster, Eric Pineda, Manal S. Fawzy, Eman A. Toraih, Emad Kandil

**Affiliations:** 1School of Medicine, Tulane University, New Orleans, LA 70112, USA; epersons@tulane.edu (E.M.P.); mherrera3@tulane.edu (M.H.); epineda@tulane.edu (E.P.); 2Ochsner Clinic Foundation, New Orleans, LA 70112, USA; mohamed.hussein@ochsner.org; 3Department of Surgery, University of Texas Southwestern Medical Center, Dallas, TX 75390, USA; dylan.pinion@utsouthwestern.edu; 4Department of Surgery, UMass Chan Medical School-Baystate, Springfield, MA 01107, USA; alyssa.webster@baystatehealth.org; 5Center for Health Research, Northern Border University, Arar 91431, Saudi Arabia; manal.darwish@nbu.edu.sa; 6Department of Surgery, School of Medicine, Tulane University, New Orleans, LA 70112, USA; ekandil@tulane.edu; 7Genetics Unit, Department of Histology and Cell Biology, Faculty of Medicine, Suez Canal University, Ismailia 41522, Egypt

**Keywords:** pediatric thyroid cancer, radioactive iodine ablation, differentiated thyroid carcinoma, SEER database, clinical outcomes, racial disparities in cancer treatment

## Abstract

The use of radioactive iodine (RAI) ablation is a common therapy among adults with differentiated thyroid cancer. However, the clinical outcomes and long-term effects of this treatment in pediatric thyroid cancer patients remain uncertain. The aim of our retrospective cohort study was to assess the risks and survival benefits of RAI in pediatric thyroid cancer patients. Using data from patients aged 21 years or younger with differentiated thyroid cancer from the SEER cancer registries, we confirmed RAI did not show any significant impact on recurrence, second malignancy, or mortality. There was no significant difference in long-term survival between pediatric patients who received RAI and those who did not. Thus, excellent clinical outcomes are achieved regardless of RAI use in differentiated pediatric thyroid cancer.

## 1. Introduction

Thyroid cancer represents the most common pediatric endocrine malignancy, with an increased incidence between 1975 and 2016 from 3.8/million (95% CI = 2.6–5.5) to 11.5/million in patients aged under 20 [1]. Accounting for approximately 90% of all thyroid cancer cases, differentiated thyroid carcinoma (DTC) poses a threat to children [2,3]. DTC in adolescents often manifests as a more aggressive or advanced disease state, frequently presenting with extrathyroidal extension, lymph node metastases, and pulmonary metastases [4,5]. While first-line treatment involves surgical intervention, radioactive iodine (RAI) ablation is almost always indicated as one of the adjuvant therapies to destroy residual cancer [6,7]. RAI has improved long-term outcomes in intermediate-risk DTC patients, but questions regarding its efficacy and potential complications in the pediatric population remain [8]. While the incidence of pediatric differentiated thyroid cancer is increasing, and metastatic disease is frequent, mortality is extremely rare, ranging between zero and one deaths per year, with better survival compared with adults who have cancer-specific 5- and 10-year survival rates between 89 and 98% [9,10]. Therefore, it is important to consider the risks and benefits of treatments like RAI in a population vulnerable to potential future morbidities.

Particularly, concerns regarding the risk of a second primary malignancy and damage to reproductive tissue are still being evaluated [11,12]. In adults, studies examining the effects of RAI on pregnancy outcomes have been contradictory, with some finding delayed childbearing, early ovarian failure, and decreased birthrates post-RAI treatment, while others found no difference in reproductive outcomes between RAI and non-RAI treatment groups [10,13,14,15,16,17,18]. In men, radioactive iodine therapy has been found to transiently decrease spermatogenesis for the first 12–18 months following treatment but ultimately resolves, with patients returning to their baseline reproductive function [18,19]. These reproductive concerns are still being researched in adults, and the same can be said for the fertility effects of RAI on pediatric patients. Few studies have examined the association between pediatric exposure to RAI and fertility, but existing evidence supports no increase in congenital abnormalities in offspring or long-term infertility [20,21]. Despite this, there is a lack of current research in this field, and the few previous studies are not enough to confidently ensure fertility after RAI treatment. Regarding the second primary malignancy, adults treated with RAI have been found to have an excess relative risk of developing solid tumors, salivary gland carcinomas, and leukemia [22,23,24,25]. However, the absolute excess risk of developing a second primary malignancy in adults was found to be approximately 0.5% over a ten-year post-treatment period and 0.44% in a cohort of children and young adults [25]. With the excess risk of second primary malignancy involved, it is important to examine the true benefits of imposing RAI on the pediatric population, especially given their low mortality due to differentiated thyroid cancer.

With these risks still undergoing further research in mind, RAI remains a recommended adjunct treatment for differentiated thyroid cancer in pediatric patients [22]. Therefore, gaining a better understanding of the trends in survival rates and potential adverse RAI outcomes in this unique patient population is essential for informed decision making and the development of personalized treatment plans. Prior population analyses revealed excellent cause-specific survival in pediatric thyroid cancer irrespective of RAI, suggesting potential overtreatment [26]. However, these studies spanned older periods, lacking detailed pathologic data, including tumor, nodule, and metastatic staging, regional extension, and tumor diameter to further risk-stratify subgroup responses. Furthermore, recent data highlight worse survival among racial and ethnic minorities, indicating a need to re-examine health disparities among the pediatric population with thyroid cancer [27]. In this investigation, we leveraged the Surveillance, Epidemiology, and End Results (SEER) database to analyze modern trends in RAI utilization among pediatric patients with differentiated thyroid cancer. Specifically, we assessed the changing landscape of practice patterns in RAI delivery over the past two decades. Moreover, we directly compared presenting features, treatment responses, survival outcomes, second malignancy rates, and health disparities between RAI and non-RAI cohorts to clarify the contemporary role of RAI in the management of pediatric thyroid cancer.

## 2. Materials and Methods

### 2.1. Data Source

We conducted a retrospective cohort study using data from the Surveillance, Epidemiology, and End Results (SEER) database (https://seer.cancer.gov/) accessed 25 January 2023, which covers approximately 34.6% of the US population [28]. Specifically, we utilized the SEER 17 and SEER Expanded Incidence (SEER 21 plus four additional registries) cohorts spanning 2000–2019. The SEER database contains deidentified patient data on cancer site, morphology, stage, first course of treatment, and survival.

### 2.2. Study Population

We included pediatric and adolescent patients (aged ≤ 22 years, according to the American Academy of Pediatrics guideline) with a single primary diagnosis of papillary thyroid carcinoma (PTC) or follicular thyroid carcinoma (FTC) in the SEER database from 2000 to 2019. Patients with thyroid cancer diagnosed only on autopsy or death certificate were excluded. To avoid confounding from other treatments, we also excluded those who received radiotherapy without surgery. Patients with other cancer diagnoses prior to their thyroid cancer were likewise excluded.

### 2.3. Variables

The variables extracted included demographic data (age, sex, race/ethnicity, and metropolitan status), tumor characteristics (histology, size, extension, and nodal and distant metastasis), treatment details (surgery, radioactive iodine, and systemic therapy), and outcomes (recurrence, second cancers, survival status/time, and cause of death). The cohort was categorized into those who received radioactive iodine ablation (RAI) and those who did not (non-RAI).

### 2.4. Outcomes

The primary outcome was thyroid cancer-specific mortality. The secondary outcomes included overall mortality, recurrence, second primary malignancies, and survival time. Recurrence was defined as the development of thyroid cancer > six months after initial diagnosis. Second primary malignancies are defined as new primary cancers that occur in a person who has had cancer in the past.

### 2.5. Statistical Analysis

Descriptive statistics were used to summarize the cohort characteristics. Chi-square and Student’s *t*-tests compared the categorical and continuous variables between the RAI and non-RAI groups. Kaplan–Meier curves displayed the survival distributions, and log-rank tests determined the group differences. Univariate and multivariate Cox regression analyses identified the factors associated with mortality. Logistic regression examined the predictors of second cancers. Two-tailed *p*-values of < 0.05 were considered statistically significant. All analyses were conducted using R version 4.0.2 (R Foundation for Statistical Computing, Vienna, Austria). The STROBE guidelines were appropriately followed.

Differential effects across race/ethnicity and tumor characteristics were evaluated in our multivariate regression models (Cox proportional hazards models for survival and logistic regression models for secondary outcomes) and not through individual subgroup analysis. This approach allowed us to assess the impact of these variables while maintaining statistical power and controlling for potential confounders, particularly given the relatively small number of events in some racial/ethnic categories and tumor subtypes. This enabled the assessment of their independent effects while adjusting for other clinical and demographic factors.

Missing data handling was performed systematically at multiple levels. For primary analytic variables, including outcomes (recurrence, survival, and second primary cancers) and key demographic/management variables, a complete case analysis was employed. Patients with missing data in these critical fields were excluded before analysis (whole paper). For TNM staging and regional extension variables, missing values were handled through listwise deletion during regression modeling. This approach was selected as these variables showed a missing completely at random (MCAR) pattern based on Little’s test.

### 2.6. Ethical Considerations

As SEER data are deidentified and publicly available, this study was exempt from institutional review board approval. We adhered to the SEER data use agreement to maintain patient privacy and confidentiality.

## 3. Results

### 3.1. Characteristics of Study Population

Figure 1 shows the workflow for patient selection from the SEER cancer registries from 2000 to 2019. A total of 5318 pediatric thyroid cancer patients were included in the analysis. The majority were female (*n* = 4441; 83.5%), White (*n* = 4443; 83.5%), living in metropolitan areas with over 1 million inhabitants (*n* = 3171; 59.7%), and over 12 years old (*n* = 4934; 92.8%). Most patients had a papillary histology type (*n* = 4918; 92.5%) (Table 1). Key differences were noted between the RAI (*n* = 2973; 55.9%) and non-RAI (*n* = 2345; 44.1%) groups. RAI patients were more likely to be younger, aged ≤ 12 years (8.6% vs. 5.5%; *p* < 0.001), have larger tumor sizes (mean diameter: 27.7 vs. 20.4 mm; *p* < 0.001), a more advanced T and N staging (T3/T4: 35.8% vs. 15.3%, *p* < 0.001; N1: 60.7% vs. 28.8%, *p* < 0.001), distant metastasis (M1: 2.7% vs. 0.9%; *p* < 0.001), and receive systemic therapy (70.6% vs. 43.2%; *p* < 0.001).

### 3.2. Incidence Trend of RAI Utility

An analysis of the national trend in RAI utilization from 2000 to 2019 revealed notable fluctuations over the two-decade span. As depicted in Figure 2, the percentage of pediatric thyroid cancer patients undergoing RAI therapy initially hovered around 57% at the start of the millennium but showed an upward trend, peaking at 65% by the end of the first decade. In subsequent years, a gradual decline was observed, with a significant drop to 38.4% by 2019.

### 3.3. Treatment Approaches and Outcomes

Almost all patients underwent cancer-directed surgery (*n* = 5222; 98.3%). Only 0.1% (*n* = 7) had recurrence, 1.4% (*n* = 77) developed second malignancies, and 0.3% (*n* = 17) had second primary thyroid cancers. No differences were observed between the RAI and non-RAI groups for these outcomes. The overall survival was 99.1%, with 0.9% (*n* = 47) mortality. No differences in survival status (0.8% vs. 0.9%; *p* = 0.82) or disease-specific deaths (0.3% vs. 0.1%; *p* = 0.41) occurred between the treatment groups.

### 3.4. Survival Analysis

Figure 3 shows multiple variables were independent predictors of worse overall survival in the multivariable analysis, including African American race having an increased risk of mortality (HR = 3.81; 95% CI = 1.07–13.49; *p* = 0.038). Undergoing cancer surgery was protective (HR = 0.08; 95% CI = 0.01–0.65; *p* = 0.019).

The Kaplan–Meier curves (Figure 4 demonstrated no significant differences in thyroid cancer-specific or overall survival between the RAI and non-RAI groups (*p* > 0.05 for both). This suggests radioactive iodine may not affect survival in pediatric thyroid cancer patients, consistent with the Cox regression analysis.

### 3.5. Risk Factors for Second Malignancies

In the multivariate analysis, systemic therapy increased the odds of second malignancies by over two-fold (OR = 2.53; 95% CI = 1.24–5.65; *p* = 0.015) compared with no therapy. RAI treatment did not impact risk (OR = 0.97; 95% CI = 0.50–1.96; *p* = 0.93) (Figure 5).

## 4. Discussion

Thyroid cancer rates among the pediatric population are on the rise, with pediatric cases reported as much more aggressive compared with adult cases despite young age being a favorable prognostic factor [5,29]. However, the increase in DTC among pediatric patients has not been accompanied by a consensus regarding best practices for RAI utilization. In this retrospective analysis of 5318 patients under the age of 22 in the SEER database, we comprehensively assess recent trends and the comparative effectiveness of RAI versus surgical interventions alone for long-term outcomes.

Notably, we demonstrated increasing administration of RAI for pediatric DTC, from 23.4% to 69.7%, over the study interval. This contrasts with recent evidence showing equivalent outcomes for low-risk pediatric patients managed without RAI compared with their RAI-treated counterparts [26]. Our analysis found that survival exceeded 99% at 5 and 10 years following diagnosis, regardless of RAI delivery. Additionally, recurrence, second primary thyroid malignancies, and disease-specific mortality were rare and found to be similar between the RAI and non-RAI groups. Collectively, our findings suggest surgical intervention facilitates excellent pediatric thyroid cancer prognoses with little influence from RAI administration. Of importance, our study did not suggest any difference in primary second malignancies, but future studies should further investigate this concern as conflicting data persist.

Although this study showed similar outcomes between the RAI and non-RAI groups, it is still important to be selective with RAI treatment to avoid overtreatment, especially in low-risk patients. A tool that could help physicians decide when RAI is necessary and beneficial is molecular profiling. Some studies show that certain mutations are associated with a better or worse response to RAI treatment [30]. For example, RET and PTC rearrangements are associated with better responses to RAI therapy, while conversely, BRAF^V600E^ mutations are linked to reduced iodine uptake and a poorer response to RAI [29]. Additionally, the thyroid differentiation scores (TDSs), which are quantitative measures of the differentiation status of thyroid tumors based on the mRNA expression levels of 16 genes involved in thyroid metabolism and function, can help predict RAI response. Tumors with a higher TDS tend to have a better response to RAI, while those with a lower TDS, which is often associated with BRAF^V600E^ mutations, have a poorer response to RAI [31,32]. These studies could be useful in determining the role of RAI treatment in pediatric and adolescent populations, given that RET/PTC is more common in pediatric papillary thyroid carcinoma and BRAF^V600E^ is less common [29]. However, the studies above and others using molecular profiling to assess RAI responses have been conducted mainly on adults. Therefore, future studies should clarify the appropriateness of RAI in pediatric and adolescent populations by incorporating molecular profiling and pathologic features beyond AJCC to personal care, as RAI indications might be more individual-dependent than previously thought. This notion is fluid with the findings of others who indicate that compared with adults, the treatment of children and young adults with DTC requires a multidisciplinary team of pediatric oncologists, endocrinologists, surgeons, pathologists, and radiologists to evaluate the multifactorial influences on responses to treatment [33].

Moreover, this study uncovered health disparities between racial groups. The analysis revealed that Black patients had a 3.8 times higher mortality risk compared with their White counterparts. Recent data on adolescent cancer agree that Black patients have a higher mortality rate in all cancer types [34]. At the same time, another study indicates that the existence of social determinants contributes to a wide disparity in pediatric thyroid cancer survival rates for non-White patients [27]. Further research must explore the mechanisms behind racial health disparities and optimal strategies to mitigate such discrepancies in outcomes.

The limitations of this investigation include a lack of data on prior radiation exposure, short-term toxicities, and post-treatment quality-of-life impacts. There may also exist variability in treatment protocols executed in the various geographical regions included in the SEER database, which is a consequence of including patients from across the entire nation. Additionally, the lack of significant mortality regardless of the treatment group in the ten years of post-treatment follow-up indicates the length of follow-up time would need to be expanded in future studies to examine long-term mortality outcomes beyond one decade. Other unmeasured confounders related to access and care preferences may explain some findings. Additionally, there is a lack of data on the impacts of RAI on quality of life, fertility, or growth in the pediatric population specifically. Future studies should investigate these outcomes to be able to make a more informed medical decision for the treatment of RAI in pediatric and adolescent populations. Furthermore, it is important to note that, as with any retrospective cohort study, selection bias is a potential risk, especially with loss to follow-up. Despite these limitations, the findings significantly contribute to the growing evidence that less-intensive treatment may provide a reasonable option in low–intermediate-risk pediatric thyroid cancer, with RAI being decided on a case-by-case basis according to individual pathologic and genomic factors.

## 5. Conclusions

In summary, surgery with close follow-up and selective use of RAI leads to exceptional clinical outcomes for most young DTC patients. However, moving forward, treatment guidelines must balance quality survival with therapy-related risks and burdens. Precision protocols pairing tumor genomic expression with tailored therapies can help prevent overtreatment, while social determinants of health should be considered when treating racial minorities. Ultimately, a shared decision-making model based on expected outcomes for individual patients and disease characteristics should dictate optimal management.

## Figures and Tables

**Figure 1 cancers-17-00107-f001:**
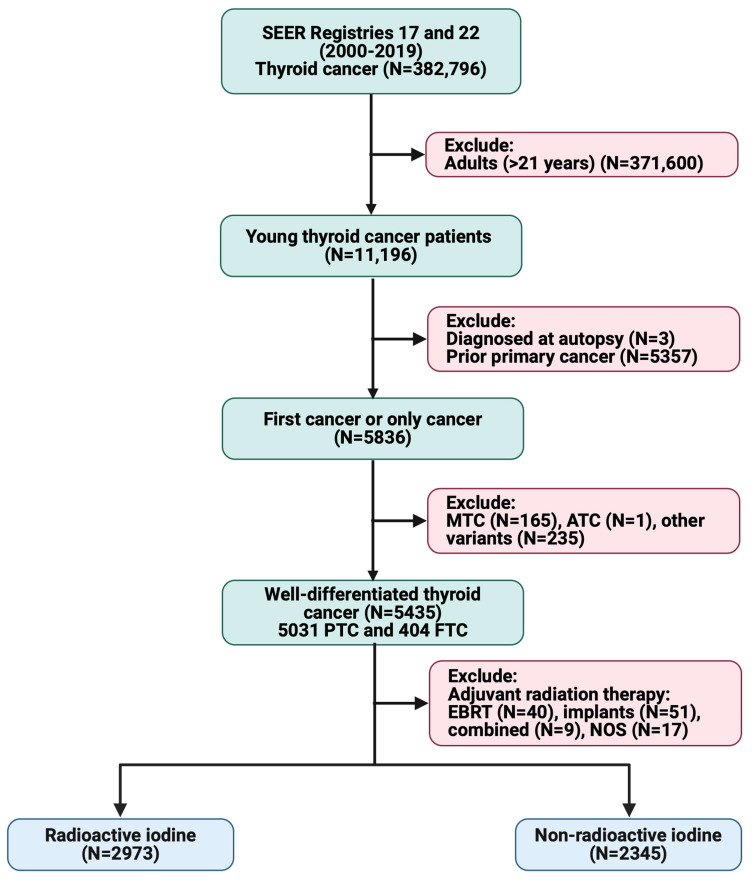
Patient selection workflow in SEER registries 17 and 22 (2000–2019). The workflow delineates the inclusion and exclusion criteria to identify the study population within the SEER database from 2000 to 2019. From a pool of 11,196 patients aged below 22, those diagnosed posthumously (*n* = 3), with other primary malignancies (*n* = 5357), or who received radiotherapy either as a single treatment or in combination (*n* = 117) were excluded. Additional exclusions were made for other thyroid cancer types (*n* = 401), culminating in a cohort of 5318 patients. Of these, 55.9% (*n* = 2973) underwent RAI therapy, while 44.1% (*n* = 2345) did not. Abbreviations: MTC (medullary thyroid cancer), ATC (anaplastic thyroid cancer), PTC (papillary thyroid cancer), FTC (follicular thyroid cancer), EBRT (external beam radiation therapy), NOS (not otherwise specified), and RAI (radioactive iodine). Data source: the Surveillance, Epidemiology, and End Results (SEER) database of cancer incidence and survival data from population-based cancer registries in the United States, maintained by the National Cancer Institute (NCI), capturing cancer incidence and survival data from population-based cancer registries in the US. The study population was obtained from 22 registries, and more variables were expanded for the same cohorts from registry 17.

**Figure 2 cancers-17-00107-f002:**
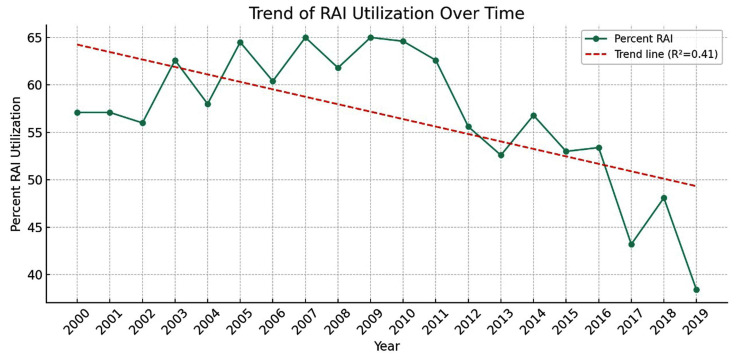
National trend of RAI utilization (2000–2019). The plot illustrates the percentage of pediatric thyroid cancer patients who received radioactive iodine (RAI) treatment each year. Data points represent the annual percentage of RAI use, while the dashed red line indicates the trend over time, with the R-squared value denoting the goodness of fit for the trend line. The trend line, while indicating a slight overall decrease (R^2^ = 0.41), suggests variations in clinical practice and possible shifts in treatment paradigms. These findings underscore the evolving landscape of RAI therapy in pediatric thyroid cancer management, reflecting changes in guidelines, practice patterns, and perhaps a growing preference for less-aggressive treatment approaches in specific subpopulations.

**Figure 3 cancers-17-00107-f003:**
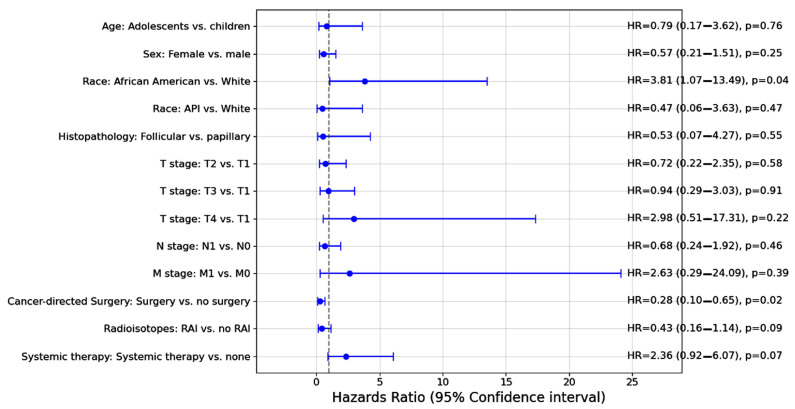
Independent predictor risk factors for overall survival. The Cox regression hazards proportional test was used. Hazard ratios and 95% confidence intervals (CIs) were reported. Adolescents’ ages were set to ≥13 years. API: Asian or Pacific Islander; RAI: radioactive iodine ablation. RAI treatment did not impact mortality risk (HR = 0.43; 95% CI = 0.16–1.14; *p* = 0.09).

**Figure 4 cancers-17-00107-f004:**
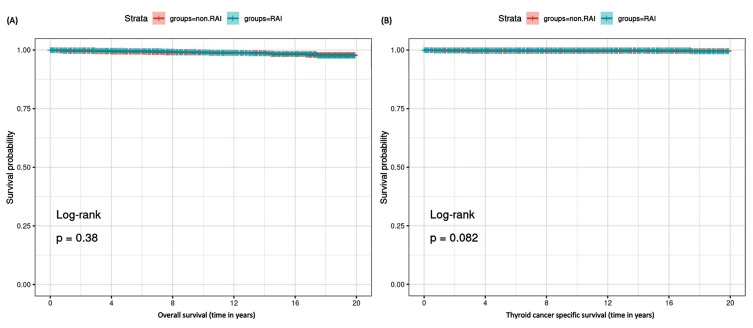
Kaplan–Meier survival curves. (**A**) Kaplan–Meier curve for overall survival. (**B**) Kaplan–Meier curve for thyroid cancer-specific survival. Log-rank test was used to compare RAI and non-RAI groups. Statistical significance was set at *p*-value < 0.05.

**Figure 5 cancers-17-00107-f005:**
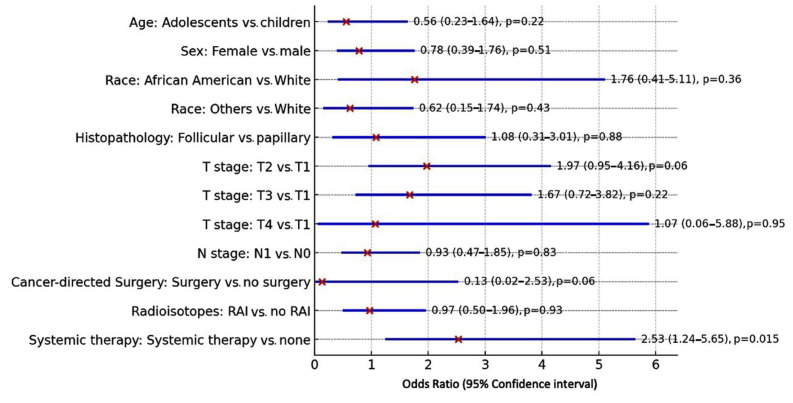
Independent predictor risk factors for second primary malignancy. A binary logistic regression analysis test was used. Odds ratios and 95% confidence intervals (CIs) were reported. Adolescents’ ages were set to ≥13 years. Other races included Asian or Pacific Islander and American Indian/Alaska Native; RAI: radioactive iodine ablation.

**Table 1 cancers-17-00107-t001:** Characteristics of the study population.

Characteristics	Levels	Total (*n* = 5318)	Non-RAI (*n* = 2345)	RAI (*n* = 2973)	*p*-Value
Demographics					
Age, years	≤12 years	384 (7.2)	129 (5.5)	255 (8.6)	**<0.001**
	>12 years	4934 (92.8)	226 (94.5)	2718 (91.4)	
Sex	Female	4441 (83.5)	1993 (85.0)	2448 (82.3)	**0.011**
	Male	877 (16.5)	352 (15.0)	525 (17.7)	
Race	White	4443 (83.5)	1948 (83.1)	2495 (83.9)	**0.005**
	American Indian/Alaska Native	47 (0.9)	15 (0.6)	32 (1.1)	
	Asian or Pacific Islander	517 (9.7)	236 (10.1)	281 (9.5)	
	Black	213 (4.0)	87 (3.7)	126 (4.2)	
	Unknown	98 (1.8)	59 (2.5)	39 (1.3)	
Hispanic/Latino	No	3899 (73.3)	1758 (75.0)	2141 (72.0)	**0.017**
	Yes	1419 (26.7)	587 (25.0)	832 (28.0)	
Metropolitan	Metropolitan > 1 M pop	3171 (59.7)	1405 (60.0)	1766 (59.5)	0.72
	Metropolitan > 250 K–1 M	1208 (22.8)	544 (23.2)	664 (22.4)	
	Metropolitan of <250 K	446 (8.4)	188 (8.0)	258 (8.7)	
	Non-metropolitan adj. to metropolitan	264 (5.0)	110 (4.7)	154 (5.2)	
	Non-metropolitan not adj. to metropolitan	229 (4.1)	98 (4.0)	131 (4.3)	
Presentation					
Histological type	Papillary	4918 (92.5)	2153 (91.8)	2765 (93.0)	0.11
	Follicular	400 (7.5)	192 (8.2)	208 (7.0)	
Max. diameter, mm	Mean (SD)	24.5 (22.2)	20.4 (16.4)	27.7 (25.5)	**<0.001**
T staging	T1	1949 (45.7)	1111 (59.1)	838 (35.1)	**<0.001**
	T2	1195 (28.0)	480 (25.5)	715 (30.0)	
	T3	983 (23.0)	265 (14.1)	718 (30.1)	
	T4	135 (3.2)	23 (1.2)	112 (4.7)	
N staging	N0	2704 (53.0)	1564 (71.2)	1140 (39.3)	**<0.001**
	N1	2394 (47.0)	632 (28.8)	1762 (60.7)	
M staging	M0	5206 (98.1)	2315 (99.1)	2891 (97.3)	**<0.001**
	M1	101 (1.9)	20 (0.9)	81 (2.7)	
Regional extension	Localized	2366 (45.3)	1461 (64.2)	905 (30.7)	**<0.001**
	Regional	2609 (49.9)	757 (33.3)	1852 (62.7)	
	Distant	252 (4.8)	57 (2.5)	195 (6.6)	
Management					
Cancer-directed surgery	Negative	92 (1.7)	91 (3.9)	1 (0.0)	**<0.001**
	Positive	5222 (98.3)	2250 (96.1)	2972 (100.0)	
Systemic therapy	Negative	1709 (41.7)	1049 (56.8)	660 (29.4)	**<0.001**
	Positive	2386 (58.3)	798 (43.2)	1588 (70.6)	
Outcomes					
Recurrence	Negative	5311 (99.9)	2341 (99.8)	2970 (99.9)	0.75
	Positive	7 (0.1)	4 (0.2)	3 (0.1)	
Second primary cancers	Negative	5241 (98.6)	2311 (98.6)	2930 (98.6)	1.0
	Positive	77 (1.4)	34 (1.4)	43 (1.4)	
SPTC, diff. histo.	Negative	5301 (99.7)	2335 (99.6)	2966 (99.8)	0.33
	Positive	17 (0.3)	10 (0.4)	7 (0.2)	
Survival status	Alive	5271 (99.1)	2323 (99.1)	2948 (99.2)	0.82
	Dead	47 (0.9)	22 (0.9)	25 (0.8)	
Cause of death	Alive	5271 (99.1)	2323 (99.1)	2948 (99.2)	0.41
	Dead, this cancer	12 (0.2)	8 (0.3)	4 (0.1)	
	Dead, other cause	35 (0.7)	14 (0.6)	21 (0.7)	

Data are presented as numbers and percentages, means and standard deviations (SDs), or medians and interquartile ranges (IQRs). Two-sided chi-square, Student’s t-, and Mann–Whitney U tests were used. Statistical significance was set at *p*-value < 0.05. RAI: radioactive iodine, SPTC: second primary thyroid cancer. The bold values indicate significance at *p*-value < 0.05.

## Data Availability

The data needed are available within this manuscript. The SEER database is available online.

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
