# Peer review of "The Impact of Radioactive Iodine on Outcomes Among Pediatric and Adolescent Thyroid Cancer Patients: A SEER Database Analysis"

_cancers, 2025, doi:10.3390/cancers17010107_

Round 1

Reviewer 1 Report

Comments and Suggestions for Authors

This is a retrospective cohort study analyzing the impact of radioactive iodine on outcomes in pediatric and adolescent thyroid cancer patients.

The title and abstract are fine, informative and succint.  The introduction section gives enough data on background issues, and the issue of potentially unnecessary RAI administration in select pediatric populations. The aims are clearly stated. 

The Methods and Materials section listed clear inclusion and exclusion criteria. The study sample is adequate. STROBE guidelines, checklists need to be cited and added. Statistics were adequate.

The main problem with defining outcomes as mortality is the fact there is almost no mortality in ten post-treatment years in well-differentiated thyroid cancer regardless of treatment modality, so the length of follow-up is the main confounding factor. 

The main group of interest here is the group of patients with metastatic disease identified at the outset of treatment, or during the immediate postoperative followup, since that is the group where surgical treatment cannot be effective. The study lists 101 (1.9) patients, with 20 (0.9) not receiving RAI, and 81 (2.7) receiving RAI. I would be curious to see more comments on the group of patients that had M1 disease and type of treatment - systemic, RAI or a combination, since that is the group most affected by this treatment modality. 

Author Response

Comment 1: This is a retrospective cohort study analyzing the impact of radioactive iodine on outcomes in pediatric and adolescent thyroid cancer patients.

Response 1: In line 18 of Simple Summary, I added “cohort” for clarification.

Comment 2: The title and abstract are fine, informative and succint.  The introduction section gives enough data on background issues, and the issue of potentially unnecessary RAI administration in select pediatric populations. The aims are clearly stated. 

Response 2: Thank you!

Comment 3: The Methods and Materials section listed clear inclusion and exclusion criteria. The study sample is adequate. STROBE guidelines, checklists need to be cited and added. Statistics were adequate.

Response 3: Thank you very much. We have added in line 142 that STROBE guidelines were followed and have attached the STROBE checklist under supplementary materials in this submission.

Comment 4: The main problem with defining outcomes as mortality is the fact there is almost no mortality in ten post-treatment years in well-differentiated thyroid cancer regardless of treatment modality, so the length of follow-up is the main confounding factor. 

Response 4: We agree and have edited to address this in lines 304-307. However, it is important to note that differentiated thyroid cancer in pediatric patients has rare mortality at baseline. 

Comment 5: The main group of interest here is the group of patients with metastatic disease identified at the outset of treatment, or during the immediate postoperative followup, since that is the group where surgical treatment cannot be effective. The study lists 101 (1.9) patients, with 20 (0.9) not receiving RAI, and 81 (2.7) receiving RAI. I would be curious to see more comments on the group of patients that had M1 disease and type of treatment - systemic, RAI or a combination, since that is the group most affected by this treatment modality. 

Response 5: Thank you for this thoughtful inquiry. We concluded any analysis with M1 patients alone (sample size of 101) would be too small for any significant findings.

Reviewer 2 Report

Comments and Suggestions for Authors

While the study is well prepared and aims to clarify the role of radioactive iodine (RAI) in pediatric thyroid cancer, it does not fully address the potential implications of emerging molecular diagnostic techniques for personalizing treatment plans. 

It would be beneficial for readers if the authors describe following.

1. Provide more detailed background on pediatric thyroid cancer and why RAI is controversial.

2. Clarify the research gap and highlight how this study uniquely contributes to the literature.

3. Justify the inclusion of patients aged up to 22 years instead of the standard pediatric cutoff of 18 years.

4. Clearly define variables like "recurrence" and "second malignancy."

5. Address whether subgroup analyses were conducted (e.g., by race or tumor type).

6. Explain how missing data were handled to ensure transparency.

7. (It is just my suggestion.) Reorganize the section for better logical flow, such as moving survival outcomes earlier.

8. Add annotations or better explanations to figures and tables, such as Figure 2 and Kaplan-Meier curves.

9. Discuss the potential for overtreatment with RAI in low-risk patients and how molecular profiling can help.

10. Compare pediatric outcomes to adult thyroid cancer outcomes for context.

11. Discuss potential selection bias due to the retrospective study design.

12. Address variability in treatment approaches across SEER regions.

13. Acknowledge the lack of data on quality-of-life impacts, fertility, or growth effects.

Author Response

Comment 1: While the study is well prepared and aims to clarify the role of radioactive iodine (RAI) in pediatric thyroid cancer, it does not fully address the potential implications of emerging molecular diagnostic techniques for personalizing treatment plans. 

Response 1: We agree that it is important to mention the implications of emerging molecular diagnostic techniques and have added more information regarding this in lines 268-284.

Comment 2: Provide more detailed background on pediatric thyroid cancer and why RAI is controversial.

Response 2: Please see lines 64-87 which were added for further context about reproductive and secondary primary malignancy risks, making RAI controversial. Additionally, lines 58-63 were added for more background on pediatric thyroid cancer mortality and why it is important to consider treatment carefully.

Comment 3: Clarify the research gap and highlight how this study uniquely contributes to the literature.

Response 3: Lines 90-97 address prior studies including older time periods, lacking detailed pathologic data, and the need for health disparities investigation.

Comment 4: Justify the inclusion of patients aged up to 22 years instead of the standard pediatric cutoff of 18 years.

Response 4: Thank you for pointing this out. The 22 year age point was a chosen according to the American Academy of Pediatrics guideline, as referenced in lines 114-115.

Comment 5: Clearly define variables like "recurrence" and "second malignancy."

Response 5: Lines 131-133 define these terms. 

Comment 6: Address whether subgroup analyses were conducted (e.g., by race or tumor type).

Response 6: Thank you for this comment. We have added further information regarding the approach to this analysis which did not include subgroup analyses and our reasoning behind this decision. This can be found on lines 143-150. 

Comment 7: Explain how missing data were handled to ensure transparency.

Response 7: We have added lines 151-158 in order to address how missing data was handled.

Comment 8: (It is just my suggestion.) Reorganize the section for better logical flow, such as moving survival outcomes earlier.

Response 8: Thank you for this suggestion. We agree and have moved survival outcomes to be earlier in the results section, now preceding risk for second primary malignancy.

Comment 9: Add annotations or better explanations to figures and tables, such as Figure 2 and Kaplan-Meier curves.

Response 9: Thank you for your comment. We feel as though the figure and table legends are comprehensive. 

Comment 10: Discuss the potential for overtreatment with RAI in low-risk patients and how molecular profiling can help.

Response 10: Lines 268-284 have been added to further address the implications of molecular profiling in future treatment planning. Thank you.

Comment 11: Compare pediatric outcomes to adult thyroid cancer outcomes for context.

Response 11: This is important to include, so thank you. We have added more information on this in lines 58-63.

Comment 12: Discuss potential selection bias due to the retrospective study design.

Response 12: We have added to the discussion in order to address this in lines 312-313. 

Comment 13: Address variability in treatment approaches across SEER regions.

Response 13: We have added lines 308-312 in order to address this.

Comment 14: Acknowledge the lack of data on quality-of-life impacts, fertility, or growth effects.

Response 14: Thank you for this comment. We agree these are important to mention and have included them in lines 308-312.